# Aspects of Sexuality in Quilombola Communities’ Young Adults in Eastern Brazilian Amazon

**DOI:** 10.3390/bs13060492

**Published:** 2023-06-12

**Authors:** Lilian Gabrielle Ramos Costa, Aline Cristina Mercês Pinheiro, Iasmin Pereira Cabral Miranda, Aline Lobato de Farias, Hilton Pereira da Silva, Fabiana de Campos Gomes, Angélica Mércia Pascon Barbosa, Thalita da Luz Costa, João Simão de Melo Neto

**Affiliations:** 1Institute of Health Sciences, Federal University of Pará (UFPA), Bethlehem 66050-160, PA, Brazil; 2Institute of Philosophy and Human Sciences, Federal University of Pará (UFPA), Bethlehem 66050-160, PA, Brazil; 3Research Unit in Genetics and Molecular Biology, Medical School of São José do Rio Preto, São Paulo 15090-000, SP, Brazil; 4Department of Physiotherapy and Occupational Therapy, School of Philosophy and Sciences, São Paulo State University (UNESP), Marília 01049-010, SP, Brazil

**Keywords:** sexual health, Amazon, black population, ethnic groups

## Abstract

Quilombola communities are descended from African slaves who escaped in resistance to imperial rule in Brazil. Today, these communities suffer from inadequate health care and health promotion programs due to socioeconomic, geographic, and political factors. This generates greater vulnerability among these groups because they have limited information about prevention to improve their quality of life. This research aimed to analyze the sexuality of young quilombola adults and the impact on their quality of life through an observational, cross-sectional, quantitative study with descriptive and inferential analyses. Our study is the first to address these issues among quilombolas in the Eastern Amazon region. The participants were 79 individuals of both sexes, aged between 18 to 35 years, belonging to seven communities in the state of Pará. The questionnaires were designed to assess sexual behavior and satisfaction, values and beliefs about sexuality, prejudice regarding sexual and gender diversity, knowledge about sexually transmitted infections (STIs), beliefs about maternity, and quality of life. Women reported greater sexual dissatisfaction and lower quality of life than men. Men reported no dysfunctions; however, they were highly prejudiced towards sexual and gender diversity. Low education negatively impacts the health of quilombola populations, as knowledge about STIs and values and beliefs influence sexual behavior, exposing individuals to diseases. The research also confirms that, both among quilombolas and other groups, factors such as sexual satisfaction, values and beliefs about reproduction, and affectivity directly influence the quality of life.

## 1. Introduction

Quilombola are populations descended from enslaved Africans who fled to the forests in resistance during the imperial period in Brazil. Today, they are predominantly concentrated in rural areas of the country. The inhabitants practice subsistence activities based on slash-and-burn agriculture, extraction of natural products from the forest, production of handicrafts for sale, traditional fishing and hunting, and raising small animals for family consumption [1,2,3]. The Brazilian government recognized the rights of quilombolas as traditional populations, and the national constitution of 1988 formally granted them ownership of their lands [4]. However, most quilombola areas are still not legalized, resulting in inadequate infrastructure and restricted access to health and education services, leaving them vulnerable and with a low quality of health and life [5,6,7,8].

Formal education in quilombola communities is challenging because there are few schools and a lack of trained teachers to promote quality education, respecting their cultural norms [9]. This is an aggravating factor for the insertion of this population in the socio-cultural and economic spheres of the country.

In addition, knowledge of sexuality is essential for general health and quality of life and covers aspects related to sexuality itself, such as sexual function and satisfaction, desire, lubrication, orgasm, and resolution, in addition to affective feelings, values, beliefs, and even knowledge about sexual expression diversities and sexually transmitted infections [10,11,12]. “Sexual health,” according to the World Health Organization [WHO] (2002) [13], is a state of physical and psychological well-being in relation to sexuality, as well as the possibility of having pleasurable and safe sexual relations, and WHO also added pleasure as a key factor in sexual health. For clarity, it is necessary to differentiate between the concepts of “sexual function” and “sexual satisfaction”. Sexual function refers to physiological changes in sexual response, and sexual satisfaction is defined as the degree of fulfillment in aspects of a person’s sexual life. Therefore, sexual satisfaction is essential for sexual health, overall well-being, and improved quality of life [14]. Limited knowledge about sexuality, sexual response, adequate hygiene, and pathologies affecting the urogenital system can trigger problems in the individual and profoundly impact sexual health.

It is worth pointing out that sexual satisfaction or dissatisfaction is influenced by culture, beliefs, and sexual and gender diversity prejudice, which play a fundamental role in the attitudes and values of individuals regarding sexuality. This fact involves not only physical pleasure but also emotional satisfaction and is a critical element for the quality of life and social relationships [15,16].

There is ample evidence of the importance of sex education for preventing early onset sexual intercourse, as well as for increasing awareness about the use of condoms and other contraceptive and preventive methods. It also reduces exposure to the risk of sexually transmitted infections (STIs), unintended pregnancy, and vulnerability to other diseases and promotes sexual health [12,17,18].

According to the Territorial Plan for Sustainable Rural Development of the Lower Tocantins (PTDRS), in quilombola communities, the education system results in unequal distribution of information. This hinders the production of knowledge and autonomy regarding health care since a learning process regarding the performance of sexuality and knowledge about the risk of STIs occurs at school [19,20].

Currently, we found no studies that address sexuality and quality of life among quilombal communities in the Equatorial Amazon. It is essential to elucidate the sexual health of populations of rural black origin to allow the implementation of tailored promotion, prevention, and treatment strategies with these communities, following the National Policy of Integral Health of the Black Population [21].

Thus, this study questions how knowledge about sexuality can influence the quality of life of young quilombolas from the Brazilian Eastern Amazon and aims to analyze the sexuality of young men and women, residents of quilombola communities in the Brazilian Eastern Amazon, regarding sexual knowledge and their beliefs, and the impact on the quality of life of these individuals.

## 2. Materials and Methods

### 2.1. Type of Study

This is an observational, cross-sectional, quantitative study with descriptive and inferential analysis.

### 2.2. Sampling

The study used simple random probability sampling with the purpose of allowing each element of the target audience the same probability of being selected. After defining the target population and identifying the sampling frame, a unique number was determined for each element, and after calculating the sample size, the random selection was made by drawing lots from the specific number of elements of the target population.

### 2.3. Participants

The sample was comprised of 79 quilombolas (women *n* = 40; men *n* = 39) residing in the communities of Acará (recognized by the Palmares Cultural Foundations law number 7.668, of 22 August 1988) [4], in Eastern Amazon. The sample size calculation was performed based on the study by Babakhani et al. (2018) [22], and a minimum size group of 74 participants was established. We compared the means of sexual function to check the difference between two different groups. For this calculation, an effect size of 0.59, α error of 0.05, and β of 0.2 were evidenced. GPower software version 3.1 [23] was used for the calculation.

### 2.4. Elegibility Criteria

Individuals of both sexes and young adults aged 18–35 years, classified by Petry (2002) [24], were included. Participants belonging to and residing in quilombola communities, recognized by Palmares Cultural Foundation for more than five years, sexually active, inactive, or asexual, who agreed to participate by signing the informed consent form (ICF), were included. Individuals were considered “sexually active” if they had any sexual activity in the three months prior to the beginning of the study. They were considered “sexually inactive” if they had no sexual activity for more than three months prior to the study [25]. An “asexual individual” was considered to have a “lack of sexual attraction or lack of interest in others” [26]. “Sexual activity” was defined as “any mutually voluntary activity with another person that involves sexual contact, whether or not intercourse or orgasm occurs” [27].

### 2.5. Data Collection

The individuals were first approached by physical therapy students trained to apply the questionnaire under the supervision of the responsible researcher. These individuals were then oriented about the research and, afterward, signed the ICF with simple and objective language. To measure sexual knowledge, the following questionnaires (all validated for Portuguese) were applied by an interviewer to optimize collection time and solve any difficulties the participants may have: 1—Socioeconomic and Sexual Profile questionnaire; 2—Female Sexual Functioning Index (FSFI); 3—International Index of Erectile Function (IIEF); 4—Index of Sexual Satisfaction (ISS); 5—Questionnaire on Values and Beliefs about Sexuality; 6—Scale of prejudice against sexual and gender diversity; 7—Questionnaire on Knowledge of Sexually Transmitted Diseases (Infections) (STD-KQ); 8—Questionnaire of Quality of Life (WHOQOL-BREF) was also applied. Questionnaires 1, 2, 4, 5, 6, 7, and 8 were assigned to females. Questionnaires 1, 3, 4, 5, 6, 7, and 8 were used for males.

The socioeconomic/demographic and sexual profile questionnaire was applied to draw a profile of the population to be studied in an objective way, and the sample was then submitted for the evaluation of quantitative aspects.

To evaluate sexual feelings and responses, the Female Sexual Functioning Index (FSFI) was used, which is a 19-item questionnaire that evaluates six domains of sexual function (desire, lubrication, arousal, orgasm, satisfaction, and pain). The lower the score, the worse the sexual function is considered [27].

The Adapted International Index of Erectile Function (IIFE) was developed exclusively for men and validated with the objective of measuring erectile function in a culturally, linguistically, and psychometrically valid way. Divided into five domains (erectile function, sexual desire, orgasm, sexual satisfaction, and general satisfaction), it has 15 questions, with a value from 1 to 5. The result consists of the sum of the values of the answers, with low values indicating poor quality of sexual life [28,29].

The Sexual Satisfaction Index (ISS) is a psychometric scale developed so that the degree of sexual satisfaction can be measured. The index is validated and contains 25 points, classified on a Likert scale from 1 (no part of the time) to 7 (always). This questionnaire addresses questions about the quality of sexual life, but in a non-invasive or non-offensive manner [30].

The Questionnaire of Values and Beliefs about Sexuality (QVC) consists of 17 items, using a Likert scale of 5 points to measure values and beliefs about female sexual functions, pregnancy, desire for motherhood, and abortion. Thus, through this, it becomes possible to better understand the factors that lead a woman to continue or terminate a pregnancy and female sexuality [31].

The sexual and gender diversity prejudice scale is a 22-item instrument, rated from 1 to 5, composed of 2 other instruments that assess prejudice against gender and another against Gender and Transgender nonconformity, and higher scores evidence greater prejudice [32].

The Sexually Transmitted Diseases (Infections) Knowledge Questionnaire (STD-KQ) consists of 28 items, and after adaptation to Brazilian Portuguese, deals with seven sexually transmitted diseases (gonorrhea, chlamydia, genital herpes, HPV, HIV/AIDS, hepatitis B, and syphilis). It is a questionnaire with answers categorized as “true”, “false”, and “don’t know”, which is easy to fill out [33,34].

The World Health Organization’s quality of life instrument (WHOQOL-BREF-26)-abbreviated version includes 26 items, is divided into four domains (physical health, psychological, social relations, and environmental) and has an overall score and a specific score per domain. It was translated and validated, and the global score refers to the average of two items that do not belong to the four domains but to the Global Health item and the Global Quality of Life item. In addition, the items are rated on a 5-point scale. Higher scores indicate better quality of life. Quality of life domain scores was calculated as usual by multiplying the domain mean score by a factor of 4, resulting in a range of 4 to 20 for each domain [35,36].

After filling out the forms, all data obtained were tabulated in Excel^®^ software 2013 for further data analysis. The research participants were not identified, maintaining total secrecy and respect for the confidentiality of information and privacy at all times and circumstances, minimized by means of numerical identification in the questionnaires. In addition, the data collected were only used for research by those responsible for the study, and the participants were instructed about the total freedom to withdraw from the research at any time if they so wished.

### 2.6. Variables

This study adopted gender as the independent variable and had as dependent variables sexual function and all its domains, sexual satisfaction and all its domains, values and beliefs and all its domains, sexual prejudice and gender diversity and all its domains, knowledge of Sexually Transmitted Diseases (STD) and all its domains, and quality of life and all its domains.

### 2.7. Statistical Analysis

For the descriptive analysis, absolute and relative frequencies (%) were used in addition to variables that direct the analysis after verifying the normality of the data through the Shapiro–Wilk test. Additionally, the mean, standard deviation (parametric data) or median, and 95% confidence interval (95% CI) (non-parametric data) were calculated. For inferential analysis, the following tests were performed: unpaired *t*-test and Mann–Whitney *U* test, for intergroup comparisons analyzed with parametric and non-parametric distributions, respectively. The relationship between variables was assessed using Pearson’s Correlation (r) test (parametric) or Spearman’s Rank (non-parametric).

## 3. Results

The characteristics of the sample are presented in Table 1. There was no significant age difference for either sex (males: 24 [23.99; 26.94] years; females: 24 [23.17; 26.28] years; U = 701; *p* = 0.4385, Mann–Whitney *U* test). Participants identified themselves as black and single. Up to two people contributed to the family income, with up to four people surviving on that income. Most participants had an active sex life. As for education, the women had completed high school, while the men had not. The women’s monthly income ranged up to 260.00 reais, and the men’s ranged from 781.00 to 1300.00 reais. Most women reported having no income of their own. Regarding the number of pregnancies, 40% of the women did not get pregnant, and 92% did not have an abortion. Regarding the number of children, the proportion did not differ significantly (female: 1 [0.63; 1.27]; male: 0 [0.44; 1.2]; U = 682.5; *p* = 0.339, Mann–Whitney *U* test).

### Sexual Satisfaction, Beliefs, and Their Correlations

Female sexual function, assessed by the FSFI, obtained a total score of 23.7 (95% CI: 19.315; 23.335). The satisfaction domain obtained the lowest score of 2.4 (95% CI: 2.148; 2.692), and the lubrication and pain domains had the highest scores of 4.5 (95% CI: 3.292; 4.328) and 4.48 (95% CI: 3.836; 5.124), respectively. The domains desire, arousal, and orgasm had respective scores of 3.6 (95% CI: 3.321; 4.029), 3.4 (95% CI: 2.880; 3.660), and 4.0 (95% CI: 3.190; 4.150).

Male individuals assessed by the IIEF were classified as having “no erectile dysfunction” with a median of 26 (95% CI: 20.04; 25.80) based on the classification of erectile dysfunction [34]. The remaining domains had the following values: satisfaction = 11 (95% CI: 7.99; 11.19); orgasm = 9 (95% CI: 6.13; 8.38); sexual desire = 9 (95% CI: 7. 99; 8.83); global satisfaction = 9 (95% CI: 8.53; 9.47); and total = 64 (95% CI: 51.46; 62.90).

Table 2 shows the difference in the level of sexual satisfaction, values, and beliefs about sexuality, level of prejudice regarding diversity and gender, knowledge about STIs, and quality of life between the sexes among the participants. The results of sexual satisfaction showed that women showed greater dissatisfaction when compared to men. Women had more negative values and beliefs about motherhood. Men indicated better quality of life in the physical, psychological, and environmental domains.

Table 3 shows the overall correlations (of both sexes) between sexual satisfaction, values and beliefs about sexuality, prejudice toward sexual and gender diversity, STD-KQ, and WHOQOL-BREF-26. Table 4 and Table 5 show the correlations divided by sex, with an analysis of male (by IIEF) and female (by FSFI) sexual function, respectively.

Almost all five domains of male sexual function (erectile function, orgasm, sexual satisfaction, sexual desire, and overall sexual function) were correlated, except sexual desire, which was not correlated with erectile function, sexual satisfaction, and orgasm. Greater knowledge of STIs was correlated with greater overall sexual function, erectile function, satisfaction, and orgasm. Greater sexual satisfaction, overall sexual function, and erectile function were correlated with lower beliefs about motherhood. Lower beliefs around motherhood were related to higher sexual satisfaction, orgasm, and overall function. Finally, better overall quality of life, including the physical aspect, was related to higher sexual desire and orgasm (Table 4).

In females, we observed that greater desire was associated with greater lubrication and less knowledge about STIs. Arousal correlated positively with lubrication, orgasm, pain, overall sexual function, and sexual satisfaction. It correlated negatively with overall quality of life and the physical, psychological, and environmental domains. Lubrication correlated with less prejudice, quality of life in social relationships, and the environment. It was associated with an increase in overall sexual function, orgasm, sexual satisfaction, pain, and knowledge about STIs. The orgasm was related to higher sexual satisfaction, pain, overall sexual function, and lower quality of life in general and in the environmental domain. Sexual satisfaction, as assessed by the FSFI, was positively related to almost all items assessed for sexual function, except desire. Increased pain was related to higher sexual function and sexual satisfaction. Finally, overall sexual function was correlated with higher scores for sexual satisfaction, abortion beliefs, and quality of life in the environmental domain.

Regardless of sex, a better quality of life (total and specific for the environmental domain) correlated with higher sexual satisfaction (Table 3). In males, there was no correlation between sexual satisfaction and other variables (Table 4). However, when analyzed by sex, the best quality of life (overall) was correlated with higher sexual satisfaction in females (Table 5).

All domains of the questionnaire on values and beliefs about sexuality were correlated. That is, high beliefs about motherhood being the primary function of women were correlated with higher beliefs in reproduction as a primary function of female sexuality, as sharing of affections, negative representations of abortion, and higher beliefs in considering pleasure as a primary function in female sexuality. Greater prejudice was correlated with greater beliefs about reproduction, affectivity, motherhood, abortion, and pleasure.

Less knowledge about STIs was related to stronger beliefs about reproduction, affectivity, and motherhood. The higher belief around reproduction was correlated with better quality of life in social relationships. The greater belief around affectivity was related to the lowest quality of life in the physical and environmental domains (Table 3).

In men, the stronger beliefs about motherhood were correlated with the highest beliefs about affectivity. Stronger beliefs around abortion and pleasure were correlated with higher beliefs about the other domains. Stronger beliefs around reproduction were correlated with a higher quality of life in the psychological and social relations domains. Greater prejudice was related to the stronger beliefs around motherhood and abortion. Less knowledge about STIs was associated with higher beliefs around reproduction and maternal pleasure (Table 4).

In women, stronger beliefs about motherhood as their primary goal were associated with higher beliefs about the need to share affections, negative representations of abortion, and stronger beliefs about pleasure as a primary function in female sexuality. Higher prejudice was related to stronger beliefs about reproduction and affectivity. Maternal beliefs about reproduction were correlated with greater pleasure, affection, and motherhood. Improved quality of life in the physical and environmental domains was associated with negative beliefs around affectivity. Less knowledge about STIs was related to stronger beliefs about motherhood (Table 5).

For prejudices against sexual and gender diversity, regardless of sex, there was a negative correlation with knowledge about STIs (Table 3). In males, higher prejudice against diversity was associated with a higher quality of life in the environment domain and lower knowledge about STIs (Table 4). For females, higher prejudice against diversity was associated with lower knowledge about STIs and lower quality of life in the psychological domain (Table 5).

Knowledge about STIs correlated negatively with quality of life in the social relations domain, regardless of gender (Table 3). For males, less knowledge about STIs was associated with higher quality of life in the psychological, social relations, and environmental domains (Table 4). However, there was no correlation of knowledge about STIs with other variables among females (Table 5).

When analyzing the population independent of sex and female sex alone, quality of life was correlated with all domains (Table 3 and Table 5).

Regarding correlation by sex, for males, more positive general perceptions of quality of life were related to worse quality of life in the physical and social relations domains. The physical domain was related to a higher quality of life in all domains. Higher quality of life in the psychological domain was associated with a higher quality of life in the social relations and the environmental domains. Therefore, the social domain correlated positively with quality of life in the environmental domain (Table 4).

When analyzing the population independent of gender and female sex alone, quality of life was correlated for all domains (Table 3 and Table 5).

Regarding the correlation by gender, for males, a more positive overall perception of quality of life was associated with worse quality of life in the physical and social relations domains. The physical domain was related to a higher quality of life in all domains. Higher quality of life in the psychological domain correlated with a higher quality of life in the social relationships and environmental domains. Therefore, the social domain correlated positively with quality of life in the environmental domain (Table 4).

## 4. Discussion

The aim of this study was to analyze the sexuality of young adults living in quilombola communities in the eastern Amazon of Brazil and the impact on their quality of life. Despite the emergence of research among quilombola communities in recent years, there remains a dearth of articles with well-developed methodologies related to sexuality. To our knowledge, this is the first study to examine sexuality among quilombolas in the eastern Amazon and quality of life. Due to the lack of data on quilombolas, we will refer to studies based on other populations to compare our findings.

According to a study by Almeida, Santos, Queiroz, and Mussi (2020) [37], most individuals in their sample classified themselves as black and had a family income of up to 780,00 Reais. In the Brazilian Institute of Geography and Statistics (2019) [38], the inequality factor in the income distribution of the black population showed that blacks are among the poorest groups in Brazil. Another study by Vieira and Monteiro (2013) [39] showed that the national population had completed elementary school. However, our study showed that males had incomplete schooling while females had completed high school. This could be because educational conditions in Quilombola have improved in recent decades, albeit to a limited extent, in quilombola education conditions. In addition, most individuals in this study reported being single, unlike in the study by Almeida et al. (2020) [37], in which the general population lived with a partner. In addition, Vieira and Monteiro (2013) [39] showed that most individuals lived in a committed relationship. These results could be related to associated with the age profile of each study.

Pechorro, Diniz, and Vieira (2009) [40] found no relationship between the dimensions of the sexual response and sexual satisfaction in a sample of 152 women (aged 26 to 70 years, Caucasians living in the city). This indicates that satisfaction and sexual function are interdependent variables in women. However, they also found a strong relationship between sexual satisfaction and sexual behavior in women, relating this finding to behaviors, such as tenderness and foreplay, and not necessarily to oral or vaginal sex itself. In contrast, our study found that, in our correlation by sex, greater sexual satisfaction was associated with greater sexual function in women. The difference in samples in the respective studies may justify this discrepancy.

In addition, sexual dysfunction affects approximately 40% of women, and the academic community is increasingly interested in this area. New diagnostic proposals have, therefore, classified these problems more comprehensively, allowing multiple aspects of women’s sexual responses to be assossed. A cross-sectional study by Pacagnella, Martinez, and Vieira (2009) [41] using the FSFI (*n* = 253) showed higher correlation rates between orgasm and arousal, satisfaction and orgasm, arousal and desire, orgasm and lubrication, and arousal and lubrication. A lower correlation was found between pain and all other domains, including satisfaction. These results are consistent with ours; however, we observed that an increase in pain was associated with an increase in function, sexual satisfaction, and orgasm.

Age is an important factor influencing the different results of the various studies. Rossi, Barbosa, and Oliveira (2016) [42] used the IIEF and found that in healthy men (*n* = 359) aged up to 59 years, 35.8% (*n* = 128) had some degree of erectile dysfunction, ranging from mild to complete dysfunction. Although our study was also conducted with healthy individuals, our population consisted of young adult men aged 18 to 35 years, and this group did not exhibit erectile dysfunction. In addition, the participants in our study reported being black, whereas Rossi et al. (2016) [42] had a majority of a white sample.

A study by Nelas, Chaves, Coutinho, and Amaral (2016) [43] among college students (*n* = 641) found high scores for beliefs and values around pleasure, followed by affectivity. Consistent with our study, low scores were found for beliefs and values around reproduction. However, all domains tend to be causally related. Thus, when beliefs are high in one domain, other domains have similar results.

In addition, the level of education and access to information is correlated to the degree of prejudice against sexual and gender diversity. Costa et al.’s study (2015) [31] found that men (*n* = 800) scored higher on prejudice, followed by religious and rural residents and people with lower education. In our study, quilombolas, both men and women, showed prejudice in sexual and gender diversity, with no significant gender differences. Consistent with Costa et al. (2015) [31], our sample consisted of rural residents who had less access to information and less education about sexuality.

The sexuality of an individual is beyond his or her genital anatomy. That is, sexuality is directly related to the sociocultural context of the individual. Thus, the socioeconomic and cultural conditions faced by young quilombolas leave them exposed and vulnerable to sexual violence. Regarding knowledge of STIs, Cardoso (2011) [44] found that the majority (84%) of the youth in his sample (*n* = 50, 18 to 24 years old) knew about STIs, while a minority (10%) were completely unaware. The commonly known diseases were gonorrhea, syphilis, and AIDS. The self-perception of the risk of contracting any STI was low, demonstrating a commonly adopted risk behavior by the disuse of condoms. Our STD-QK results showed that the mean score for men was 7.59 ± 4.375 and 6.98 ± 4.605 for women, demonstrating limited knowledge about STIs. These results indicate a higher risk of contracting STIs among quilombola populations due to a lack of access to prevention information, as well as a lack of health education and implementation of health policies in these communities [45].

Quality of life is “an individual’s perception of their position in the context of the culture and value system in which they live, and in relation to their goals, expectations, standards, and concerns”, and involves dimensions such as physical and psychological health, independence, social relationships, environment, and spiritual standards [46].

Almeida-Brasil (2017) [47] assessed the quality of life of individuals (*n* = 930) seen at UBS in Belo Horizonte using the WHOQOL-BREF and concluded that the score in the social relations domain was higher, with lower means in the environment domain. The negative quality of life results are related to “worse conditions of health, housing, education, and income, in addition to problems in social relationships and psychological conditions” [46].

Horta, Cruz, and Carvalho (2019) [48] studied the quality of life of African refugees (*n* = 31), and 48.38% of the participants rated their quality of life as “neither good nor bad”, followed by 29.03% who rated it as “bad”. Regarding satisfaction with health, 35.2% described themselves as “dissatisfied.” However, while the quilombola population has less access to the factors mentioned by Almeida-Brasil et al. (2017) [47], the perception of quality of life was not lower in our sample. In addition, other factors, such as sexual function and beliefs, correlated with a higher quality of life scores in the setting.

### Limitations of the Study

Due to the SARS-CoV-2 pandemic, we had difficulty entering communities and collecting data because of the risk of exposure to researchers and respondents, which limited the number of interviews. In this sample, although there were individuals who reported having a high school education, the application of the questionnaires showed gaps that were inconsistent with this indication, suggesting possible significant educational deficits in this population in general. Another limiting factor of our study was the classification as sexually inactive in those whose sexual activity had occurred more than 3 months ago without taking into consideration autoerotic activity. In addition, we were able to find limited indexed articles with well-developed methods and a similar research design that addressed sexuality and quilombola communities to add to the discussion.

## 5. Conclusions

In the communities included in this study, we found that women had greater sexual dissatisfaction and poorer quality of life than men. Most men did not have sexual dysfunction. In addition, both men and women had similar values and beliefs about reproduction, affectivity, abortion, and pleasure. However, women had negative attitudes toward motherhood because they felt pressured to become mothers. Men had more biases related to sexual diversity and gender. This can be attributed to the cultural expectations associated with historical gender roles of masculinity in Brazil, in which heterosexuality and marked masculinity are expected, and the sexualized being with natural desires is disregarded. This again underscores the importance of discussing forms of identity that defy the heterosexual norm. Moreover, many people in Brazil fear that relationships with people of the same sex are acceptable or forbidden. Same-sex intercourse is acceptable only in secret, and those involved publicly assume stable heterosexual relationships [49].

We also conclude that educational aspects may influence the sexuality of the quilombola population, as limited knowledge about STIs and values and beliefs make them more vulnerable. Therefore, this study highlights the importance of implementing public policies to promote health and education for ethnic and racial equity and to ensure the active participation of quilombos to improve their quality of life [9,21].

This study also confirms that among quilombos, as among other groups, factors, such as sexual satisfaction, reproductive values and beliefs, and affectivity, directly influence quality of life. Our study is the first to address these issues among quilombolas in the Amazon region; therefore, further research is urgently needed among these populations in Brazil.

## Figures and Tables

**Table 1 behavsci-13-00492-t001:** Characterization of the sample.

	Women *n* = 40 (%)	Men *n* = 39 (%)	Total *n* = 79 (%)
Ethnicity			
White	1 (2.50)	-	1 (1.26)
Black	26 (65.00)	26 (66.67)	52 (65.82)
Brown	13 (32.50)	12 (30.77)	25 (31.64)
Indigenous	-	1 (2.56)	1 (1.26)
Marital Status			
Single	25 (62.50)	28 (71.79)	53 (67.10)
Married	15 (37.50)	11 (28.21)	26 (32.90)
Scholarity			
Incomplete Elementary School	8 (20.00)	10 (25.64)	18 (22.78)
Complete Elementary School	1 (2.50)	1 (2.56)	2 (2.53)
Incomplete High School	6 (15.00)	14 (35.90)	20 (25.31)
Complete High School	21 (52.50)	12 (30.77)	33 (41.77)
Incomplete Higher Education	4 (10.00)	2 (5.13)	6 (7.59)
Family income			
Up to 260.00	15 (37.50)	5 (12.82)	20 (25.31)
From 261.00 to 780.00	13 (32.50)	6 (15.38)	19 (24.05)
From 781.00 to 1300.00	9 (22.50)	15 (38.46)	24 (30.37)
From 1301.00 to 1820.00	2 (5.00)	5 (12.82)	7 (8.86)
From 1821.00 to 2600.00	1 (2.50)	5 (12.82)	6 (7.59)
From 2601.00 to 3900.00	-	3 (7.69)	3 (3.79)
N. of people that contribute to the income of the house			
One	4 (10.26)	9 (23.08)	13 (16.45)
Two	21 (53.85)	13 (33.33)	34 (43.03)
Three	6 (15.38)	9 (23.08)	15 (18.98)
Four	6 (15.38)	3 (7.69)	9 (11.39)
Five	-	3 (7.69)	3 (3.79)
More than five	2 (5.13)	2 (5.13)	4 (5.06)
N. of people supported by the house income			
One	1 (2.50)	2 (5.13)	3 (3.79)
Two	7 (17.50)	5 (12.82)	12 (15.18)
Three	9 (22.50)	10 (25.64)	19 (24.05)
Four	14 (35.00)	10 (25.64)	24 (30.37)
Five	4 (10.00)	3 (7.69)	7 (8.86)
More than five	5 (12.50)	9 (23.08)	14 (17.72)
Individual monthly paid activity			
I do not own	18 (45.00)	4 (10.26)	22 (27.84)
Up to 260.00	17 (42.50)	11 (28.21)	28 (35.44)
From 261.00 to 780.00	2 (5.00)	8 (20.51)	10 (12.65)
From 781.00 to 1300.00	3 (7.50)	12 (30.77)	15 (18.98)
From 1301.00 to 1820.00	-	3 (7.69)	3 (3.79)
From 1821.00 to 2600.00	-	1 (2.56)	1 (1.26)
Sexual life			
Active	34 (85.00)	35 (89.74)	69 (87.34)
Assexual	-	-	-
Inactive	6 (15.00)	4 (10.26)	10 (12.65)
Pregnancy			
One	9 (22.50)	-	9 (11.39)
Two	12 (30.00)	-	12 (15.18)
Three	2 (5.00)	-	2 (2.53)
Above four	1 (2.50)	-	1 (1.26)
Zero	16 (40.00)	-	16 (20.25)
Abortion			
Yes	2 (5.00)	-	2 (2.53)
No	38 (95.00)	-	37 (48.09)

**Table 2 behavsci-13-00492-t002:** Differences in level of sexual satisfaction; values and beliefs about sexuality; level of prejudice in relation to sexual and gender diversity; knowledge of sexually transmitted disease (Infections) (STD-KQ); and quality of life between the sexes in these individuals.

Instruments	Men (*n* = 39)	Women (*n* = 40)	T or U	*p*
Sexual Satisfaction	17.45 ± 7.16	21.37 ± 10.29	−1.889	0.0315 *
Values and Beliefs (Reproduction)	8.0 (7.99; 9.70)	8.0 (7.13; 8.47)	626.500	0.124
Values and Beliefs (Affectivity)	12 (11.8; 12.25)	12(10.95; 11.90)	707.000	0.454
Values and Beliefs (Maternity)	16.36 ± 3.752	14.43 ± 3.706	2.305	0.012 *
Values and Beliefs (Abortion)	12 (11.07; 12.57)	12 (11.02; 12.13)	758.500	0.828
Values and Beliefs (Pleasure)	12 (12.02; 13.21)	12 (11.67; 12.68)	664.000	0.218
Prejudice	49.77 ± 14.202	44.60 ± 16.311	1.501	0.069
STD-KQ	7.59 ± 4.375	6.98 ± 4.605	0.608	0.2725
Quality of life (total score)	4 (3.7686; 4.3340)	4 (3.5828; 4.1095)	632.500	0.182
Quality of life (physical domain)	4.2857 (4.1533; 4.4547)	4 (3.7492; 4.1213)	481.000	0.005 *
Quality of life (psychological domain)	4.1667 (4.1100; 4.3600)	4 (3.7827; 4.106)	499.500	0.009 *
Quality of life (social relationships)	4 (3.9840; 4.3579)	4 (4.0342; 4.3846)	792.000	0.747
Quality of life (environment)	3.55 ± 0.58	3.34 ± 0.52	1.693	0.0475 *

* *p* < 0.05. T or U: Test t (T) ou teste de Mann–Whitney (U). Questionnaire on Knowledge of Sexually Transmitted Diseases (Infections) (STD-KQ).

**Table 3 behavsci-13-00492-t003:** Correlation between variables regardless of sexes.

	Sexual Satisfaction	Values and Beliefs (Reproduction)	Values and Beliefs (Affectivity)	Values and Beliefs (Maternity)	Values and Beliefs (Abortion)	Values and Beliefs (Pleasure)	Prejudice	STD-KQ	Quality of Life (Total Score)	Quality of Life (Physical Domain)	Quality of Life (Psychological Domain)	Quality of Life (Social Relationships)
Values and Beliefs (Reproduction)	P = 0.120 *p* = 0.160	-	-	-	-	-	-	-	-	-	-	-
Values and Beliefs (Affectivity)	P = −0.035 *p* = 0.387	P = 0.245 *p* = 0.014 *	-	-	-	-	-	-	-	-	-	-
Values and Beliefs (Maternity)	P = 0.146 *p* = 0.112	P = 0.240 *p* = 0.016 *	P = 0.440 *p* = 0.000 **	-	-	-	-	-	-	-	-	-
Values and Beliefs (Abortion)	P = 0.070 *p* = 0.282	P = 0.282 *p* = 0.006 *	P = 0.526 *p* = 0.000 **	P = 0.462 *p* = 0.000 **	-	-	-	-	-	-	-	-
Values and Beliefs (Pleasure)	P = 0.100 *p* = 0.204	P = 0.482 *p* = 0.000 **	P = 0.372 *p* = 0.000 **	P = 0.349 *p* = 0.001 **	P = 0.432 *p* = 0.000 **	-	-	-	-	-	-	-
Prejudice	P = 0.018 *p* = 0.441	P = 0.300 *p* = 0.004 *	P = 0.227 *p* = 0.022 *	P = 0.528 *p* = 0.000 **	P = 0.332 *p* = 0.01 *	P = 0.309 *p* = 0.003 *	-	-	-	-	-	-
STD-KQ	P = 0.006 *p* = 0.481	P = −0.250 *p* = 0.010 *	P = −0.193 *p* = 0.044 *	P = −0.302 *p* = 0.003 *	P = −0.115 *p* = 0.154	P = −0.076 *p* = 0.253	P = −0.288 *p* = 0.005 *	-	-	-	-	-
Quality of life (total score)	P = −0.308 *p* = 0.005 *	P = 0.135 *p* = 0.120	P = −0.008 *p* = 0.473	P = 0.088 *p* = 0.221	P = 0.047 *p* = 0.342	P = 0.040 *p* = 0.364	P = −0.027 *p* = 0.407	*P* = 0.089 *p* = 0.219	-	-	-	-
Quality of life (physical domain)	S = −0.071 *p* = 0.279	S = 0.113 *p* = 0.163	S = −0.218 *p* = 0.027*	S = 0.035 *p* = 0.380	S = −0.003 *p* = 0.489	S = 0.018 *p* = 0.437	S = 0.000 *p* = 0.499	S = 0.092 *p* = 0.212	S = 0.420 *p* = 0.000 **	-	-	-
Quality of life (psychological domain)	S = −0.115 *p* = 0.170	S = 0.140 *p* = 0.111	S = −0.024 *p* = 0.419	S = −0.092 *p* = 0.212	S = 0.123 *p* = 0.142	S = −0.023 *p* = 0.421	S = −0.036 *p* = 0.377	S = −0.129 *p* = 0.130	S = 0.220 *p* = 0.026 *	S = 0.556 *p* = 0.000 **	-	-
Quality of life (social relationships)	P = −0.101 *p* = 0.201	P = 0.232 *p* = 0.021 *	P = −0.035 *p* = 0.380	P = 0.027 *p* = 0.408	P = 0.025 *p* = 0.414	P = −0.151 *p* = 0.094	P = 0.184 *p* = 0.053	P = −0.313 *p* = 0.003 *	P = 0.217 *p* = 0.028 *	P = 0.451 *p* = 0.000 **	P = 0.397 *p* = 0.000 **	-
Quality of life (environment)	P = −0.198 *p* = 0.049 *	P = 0.065 *p* = 0.286	P = −0.239 *p* = 0.018 *	P = 0.000 *p* = 0.498	P = −0.107 *p* = 0.175	P = −0.144 *p* = 0.104	P = 0.132 *p* = 0.124	P = −0.140 *p* = 0.111	P = 0.329 *p* = 0.002 *	P = 0.415 *p* = 0.000 **	P = 0.322 *p* = 0.002 *	P = 0.432 *p* = 0.000 **

P: r of Pearson; S: r of Spearman; *p*: <0.05. * Low correlation: P or S < 0.33. ** Moderate correlation: P or S 0.34 to 0.66. Questionnaire on Knowledge of Sexually Transmitted Diseases (STD-KQ).

**Table 4 behavsci-13-00492-t004:** Correlation between the variables in males.

	IIEF (Erectile Function)	IIEF (Sexual Satisfaction)	IIEF (Orgasm)	IIEF (Sexual Desire)	IIEF (General Satisfaction)	IIEF (Total)	Sexual Satisfaction	Values and Beliefs (Reproduction)	Values and Beliefs (Affectivity)	Values and Beliefs (Maternity)	Values and Beliefs (Abortion)	Values and Beliefs (Pleasure)	Prejudice	STD-KQ	Quality of Life (Total Score)	Quality of Life (Physical Domain)	Quality of Life (Psychological Domain)	Quality of Life (Social Relationships)
IIEF (Sexual Satisfaction)	S = 0.548 *p* = 0.000	-	-	-	-	-	-	-	-	-	-	-	-	-	-	-	-	-
IIEF (Orgasm)	S = 0.657 *p* = 0.000 **	S = 0.672 *p* = 0.000 ***	-	-	-	-	-	-	-	-	-	-	-	-	-	-	-	-
IIEF (Sexual Desire)	S = 0.164 *p* = 0.160	S = 0.257 *p* = 0.057	S = 0.256 *p* = 0.058	-	-	-	-	-	-	-	-	-	-	-	-	-	-	-
IIEF (General Satisfaction)	S = 0.379 *p* = 0.009 **	S = 0.343 *p* = 0.016 **	S = 0.534 *p* = 0.000 **	S = 0.543 *p* = 0.000 **	-	-	-	-	-	-	-	-	-	-	-	-	-	-
IIEF (Total)	S = 0.804 *p* = 0.000 ***	S = 0.828 *p* = 0.000 ***	S = 0.860 *p* = 0.000 ***	S = 0.459 *p* = 0.002 **	S = 0.619 *p* = 0.000 **	-	-	-	-	-	-	-	-	-	-	-	-	-
Sexual Satisfaction	P = 0.133 *p* = 0.210	P = 0.247 *p* = 0.065	P = 0.222 *p* = 0.87	P = 0.108 *p* = 0.256	P = −0.017 *p* = 0.459	P = 0.186 *p* = 0.128	-	-	-	-	-	-	-	-	-	-	-	-
Values and Beliefs (Reproduction)	S = −0.125 *p* = 0.224	S = −0.208 *p* = 0.101	S = −0.040 *p* = 0.406	S = 0.162 *p* = 0.162	S = −0.157 *p* = 0.170	S = −0.127 *p* = 0.221	S = 0.104 *p* = 0.265	-	-	-	-	-	-	-	-	-	-	-
Values and Beliefs (Affectivity)	S = −0.231 *p* = 0.078	S = −0.426 *p* = 0.003 **	S = −0.275 *p* = 0.045 *	S = −0.001 *p* = 0.497	S = −0.025 *p* = 0.441	S = −0.302 *p* = 0.031 *	S = −0.029 *p* = 0.431	S = 0.136 *p* = 0.205	-	-	-	-	-	-	-	-	-	-
Values and Beliefs (Maternity)	P = −0.388 *p* = 0.007 **	P = −0.365 *p* = 0.011 **	P = −0.263 *p* = 0.053	P = −0.140 *p* = 0.198	P = −0.077 *p* = 0.320	P = −0.366 *p* = 0.011 **	P = 0.126 *p* = 0.222	P = 0.176 *p* = 0.141	P = 0.506 *p* = 0.001 **	-	-	-	-	-	-	-	-	-
Values and Beliefs (Abortion)	S = −0.167 *p* = 01155	S = −0.156 *p* = 0.171	S = −0.048 *p* = 0.387	S = −0.075 *p* = 0.325	S = −0.116 *p* = 0.240	S = −0.139 *p* = 0.200	S = 0.157 *p* = 0.170	S = 0.387 *p* = 0.007 **	S = 0.484 *p* = 0.001 **	S = 0.521 *p* = 0.000 **	-	-	-	-	-	-	-	-
Values and Beliefs (Pleasure)	S = −0.145 *p* = 0.190	S = −0.302 *p* = 0.031 *	S = −0.197 *p* = 0.114	S = −0.075 *p* = 0.325	S = −0.258 *p* = 0.056	S = −0.248 *p* = 0.064	S = −0.007 *p* = 0.482	S = 0.277 *p* = 0.044 *	S = 0.431 *p* = 0.003 **	S = 0.343 *p* = 0.016 **	S = 0.613 *p* = 0.000 **	-	-	-	-	-	-	-
Prejudice	P = −0.091 *p* = 0.291	P = −0.061 *p* = 0.357	P = −0.072 *p* = 0.333	P = 0.175 *p* = 0.144	P = 0.254 *p* = 0.059	P = −0.043 *p* = 0.397	P = 0.116 *p* = 0.242	P = 0.232 *p* = 0.078	P = −0.008 *p* = 0.480	P = 0.313 *p* = 0.026 *	P = 0.330 *p* = 0.020 *	P = 0.213 *p* = 0.097	-	-	-	-	-	-
STD-KQ	P = 0.382 *p* = 0.008 **	P = 0.316 *p* = 0.025 *	P = 0.269 *p* = 0.049 *	P = −0.016 *p* = 0.462	P = 0.116 *p* = 0.241	P = 0.342 *p* = 0.017 **	P = 0.208 *p* = 0.102	P = −0.342 *p* = 0.017 **	P = −0.256 *p* = 0.058	P = −0.351 *p* = 0.014 **	P = −0.151 *p* = 0.179	P = −0.111 *p* = 0.250	P = −0.311 *p* = 0.027*	-	-	-	-	-
Quality of life (total score)	S = 0.094 *p* = 0.284	S = −0.097 *p* = 0.278	S = 0.277 *p* = 0.044 *	S = −0.086 *p* = 0.301	S = 0.079 *p* = 0.316	S = 0.098 *p* = 0.276	S = −0.190 *p* = 0.124	S = 0.210 *p* = 0.100	S = 0.013 *p* = 0.470	S = 0.127 *p* = 0.221	S = 0.070 *p* = 0.336	S = 0.132 *p* = 0.211	S = −0.012 *p* = 0.471	S = −0.067 *p* = 0.343	-	-	-	-
Quality of life (physical domain)	S = 0.110 *p* = 0.253	S = 0.260 *p* = 0.055	S = 0.327 *p* = 0.021 *	S = 0.037 *p* = 0.410 **	S = −0.042 *p* = 0.399	S = 0.255 *p* = 0.059	S = −0.001 *p* = 0.497	S = 0.012 *p* = 0.472	S = −0.166 *p* = 0.157	S = −0.008 *p* = 0.480	S = 0.072 *p* = 0.331	S = −0.057 *p* = 0.366	S = −0.003 *p* = 0.493	S = 0.175 *p* = 0.143	S = 0.479 *p* = 0.001 **	-	-	-
Quality of life (psychological domain)	S = −0.102 *p* = 0.269	S = 0.117 *p* = 0.240	S = −0.023 *p* = 0.446	S = 0.252 *p* = 0.061	S = −0.155 *p* = 0.174	S = 0.032 *p* = 0.424	S = 0.085 *p* = 0.304	S = 0.379 *p* = 0.009 **	S = 0.015 *p* = 0.464	S = −0.150 *p* = 0.181	S = 0.097 *p* = 0.279	S = −0.029 *p* = 0.431	S = 0.122 *p* = 0.229	S = −0.362 *p* = 0.012 **	S = 0.015 *p* = 0.464	S = 0.409 *p* = 0.005 **	-	-
Quality of life (social relationships)	S = 0.040 *p* = 0.405	S = 0.002 *p* = 0.496	S = 0.159 *p* = 0.167	S = 0.228 *p* = 0.081	S = 0.001 *p* = 0.498	S = 0.098 *p* = 0.276	S = −0.138 *p* = 0.201	S = 0.429 *p* = 0.003 **	S = −0.146 *p* = 0.188	S = 0.003 *p* = 0.492	S = 0.047 *p* = 0.389	S = −0.240 *p* = 0.071	S = 0.192 *p* = 0.120	S = −0.389 *p* = 0.007 **	S = 0.296 *p* = 0.034 *	S = 0.543 *p* = 0.000 **	S = 0.556 *p* = 0.000 **	-
Quality of life (environment)	P = −0.166 *p* = 0.156	P = −0.237 *p* = 0.073	P = −0.133 *p* = 0.210	P = −0.229 *p* = 0.080	P = 0.012 *p* = 0.472	P = −0.192 *p* = 0.121	P = −0.111 *p* = 0.251	P = 0.147 *p* = 0.185	P = −0.243 *p* = 0.068	P = 0.051 *p* = 0.380	P = −0.043 *p* = 0.398	P = −0.161 *p* = 0.163	P = 0.271 *p* = 0.048 *	P = −0.273 *p* = 0.046 *	P = 0.237 *p* = 0.073	P = 0.352 *p* = 0.014 *	P = 0.282 *p* = 0.041 *	P = 0.584 *p* = 0.000 **

P: r of Pearson; S: r of Spearman; *p*: <0.05. * Low correlation: P or S < 0.33. ** Moderate correlation: P or S 0.34 to 0.66. *** High correlation: P or S > 0.66. Questionnaire on Knowledge of Sexually Transmitted Diseases (STD-KQ).

**Table 5 behavsci-13-00492-t005:** Correlation between the variables in females.

	FSFI (Desire)	FSFI (Excitement)	FSFI (Lubrication)	FSFI (Orgasm)	FSFI (Satisfaction)	FSFI (Pain)	FSFI (Total)	Sexual Satisfaction	Values and Beliefs (Reproduction)	Values and Beliefs (Affectivity)	Values and Beliefs (Maternity)	Values and Beliefs (Abortion)	Values and Beliefs (Pleasure)	Prejudice	STD-KQ	Quality of Life (Total Score)	Quality of Life (Physical Domain)	Quality of Life (Psychological Domain)	Quality of Life (Social Relationships)
FSFI (Excitement)	S = −0.160 *p* = 0.162	-	-	-	-	-	-	-	-	-	-	-	-	-	-	-	-	-	-
FSFI (Lubrication)	S = −0.296 *p* = 0.032 *	S = 0.721 *p* = 0.000 ***	-	-	-	-	-	-	-	-	-	-	-	-	-	-	-	-	-
FSFI (Orgasm)	S = −0.197 *p* = 0.111	S = 0.629 *p* = 0.000 **	S = 0.687 *p* = 0.000 ***	-	-	-	-	-	-	-	-	-	-	-	-	-	-	-	-
FSFI (Satisfaction)	S = −0.120 *p* = 0.231	S = 0.567 *p* = 0.000 **	S = 0.584 *p* = 0.000 **	S = 0.769 *p* = 0.000 ***	-	-	-	-	-	-	-	-	-	-	-	-	-	-	-
FSFI (Pain)	S = −0.212 *p* = 0.094	S = 0.384 *p* = 0.007 **	S = 0.505 *p* = 0.000 **	S = 0.508 *p* = 0.000 **	S = 0.574 *p* = 0.000 **	-	-	-	-	-	-	-	-	-	-	-	-	-	-
FSFI (Total)	S = 0.024 *p* = 0.441	S = 0.690 *p* = 0.000 ***	S = 0.720 *p* = 0.000 ***	S = 0.816 *p* = 0.000 ***	S = 0.793 *p* = 0.000 ***	S = 0.699 *p* = 0.000 ***	-	-	-	-	-	-	-	-	-	-	-	-	-
Sexual Satisfaction	P = −0.021 *p* = 0.454	P = 0.437 *p* = 0.006 **	P = 0.502 *p* = 0.002 **	P = 0.500 *p* = 0.002 **	P = 0.487 *p* = 0.002 **	P = 0.297 *p* = 0.050 *	P = 0.501 *p* = 0.002 **	-	-	-	-	-	-	-	-	-	-	-	-
Values and Beliefs (Reproduction)	S = −0.018 *p* = 0.456	S = 0.054 *p* = 0.370	S = −0.042 *p* = 0.399	S = 0.063 *p* = 0.349	S = 0.064 *p* = 0.348	S = −0.182 *p* = 0.130	S = 0.068 *p* = 0.339	S = 0.295 *p* = 0.051	-	-	-	-	-	-	-	-	-	-	-
Values and Beliefs (Affectivity)	S = −0.101 *p* = 0.267	S = 0.067 *p* = 0.341	S = −0.090 *p* = 0.290	S = 0.062 *p* = 0.351	S = −0.089 *p* = 0.292	S = −0.060 *p* = 0.357	S = 0.026 *p* = 0.436	S = −0.161 *p* = 0.190	S = 0.172 *p* = 0.144	-	-	-	-	-	-	-	-	-	-
Values and Beliefs (Maternity)	P = −0.091 *p* = 0.289	P = 0.144 *p* = 0.187	P = −0.001 *p* = 0.497	P = 0.114 *p* = 0.241	P = 0.082 *p* = 0.308	P = 0.101 *p* = 0.267	P = 0.087 *p* = 0.306	P = 0.259 *p* = 0.076	P = 0.223 *p* = 0.083	P = 0.359 *p* = 0.012 **	-	-	-	-	-	-	-	-	-
Values and Beliefs (Abortion)	S = −0.033 *p* = 0.419	S = 0.255 *p* = 0.056	S = 0.085 *p* = 0.302	S = 0.167 *p* = 0.151	S = 0.345 *p* = 0.015 **	S = 0.232 *p* = 0.075	S = 0.280 *p* = 0.040 *	S = −0.068 *p* = 0.356	S = 0.078 *p* = 0.316	S = 0.418 *p* = 0.004 **	S = 0.379 *p* = 0.008 **	-	-	-	-	-	-	-	-
Values and Beliefs (Pleasure)	S = 0.083 *p* = 0.305	S = 0.019 *p* = 0.453	S = −0.001 *p* = 0.497	S = −0.009 *p* = 0.479	S = −0.002 *p* = 0.494	S = −0.134 *p* = 0.205	S = 0.087 *p* = 0.297	S = 0.243 *p* = 0.090	S = 0.655 *p* = 0.000 **	S = 0.383 *p* = 0.007 **	S = 0.319 *p* = 0.022 *	S = 0.253 *p* = 0.058	-	-	-	-	-	-	-
Prejudice	P = −0.023 *p* = 0.444	P = −0.180 *p* = 0.133	P = −0.285 *p* = 0.037 *	P = −0.146 *p* = 0.184	P = −0.062 *p* = 0.353	P = −0.119 *p* = 0.232	P = −0.194 *p* = 0.115	P = −0.022 *p* = 0.452	P = 0.327 *p* = 0.020 *	P = 0.455 *p* = 0.002 **	P = 0.681 *p* = 0.000 ***	P = 0.339 *p* = 0.016	P = 0.380 *p* = 0.008 **	-	-	-	-	-	-
STD-KQ	P = −0.390 *p* = 0.006 **	P = 0.140 *p* = 0.195	P = 0.275 *p* = 0.043 *	P = 0.066 *p* = 0.344	P = −0.018 *p* = 0.456	P = 0.173 *p* = 0.143	P = 0.098 *p* = 0.274	P = −0.116 *p* = 0.264	P = −0.219 *p* = 0.087	P = −0.139 *p* = 0.195	P= −0.312 *p* = 0.025 *	P = −0.08 *p* = 0.295	P = −0.059 *p* = 0.358	P = −0.300 *p* = 0.030 *	-	-	-	-	-
Quality of life (total score)	S = 0.00 *p* = 0.478	S = −0.287 *p* = 0.038 *	S = −0.183 *p* = 0.133	S = −0.320 *p* = 0.024 *	S = −0.249 *p* = 0.063	S = −0.007 *p* = 0.482	S = −0.262 *p* = 0.053	S = −0.341 *p* = 0.028 **	S = −0.032 *p* = 0.423	S = −0.213 *p* = 0.096	S = 0.008 *p* = 0.480	S = −0.022 *p* = 0.447	S = 0.01 *p* = 0.468	S = −0.019 *p* = 0.453	S = 0.194 *p* = 0.118	-	-	-	-
Quality of life (physical domain)	S = 0.047 *p* = 0.388	S = −0.324 *p* = 0.022 *	S = −0.175 *p* = 0.143	S = −0.252 *p* = 0.061	S = 0.008 *p* = 0.480	S = −0.002 *p* = 0.496	S = −0.169 *p* = 0.152	S = 0.004 *p* = 0.492	S = 0.090 *p* = 0.293	S = −0.392 *p* = 0.007 **	S = −0.086 *p* = 0.300	S = −0.125 *p* = 0.225	S = −0.032 *p* = 0.424	S = −0.161 *p* = 0.164	S = −0.02 *p* = 0.432	S = 0.344 *p* = 0.016 **	-	-	-
Quality of life (psychological domain)	S = 0.086 *p* = 0.301	S = −0.314 *p* = 0.026 *	S = −0.202 *p* = 0.109	S = −0.265 *p* = 0.052	S = −0.068 *p* = 0.340	S = −0.047 *p* = 0.389	S = −0.241 *p* = 0069	S = −0.148 *p* = 0.210	S = −0.269 *p* = 0.049	S = −0.174 *p* = 0.144	S = −0.245 *p* = 0.066	S = 0.109 *p* = 0.254	S = −0.220 *p* = 0.090	S = −0.308 *p* = 0.028 *	S = 0.056 *p* = 0.368	S = 0.393 *p* = 0.007 **	S = 0.618 *p* = 0.000 **	-	-
Quality of life (social relationships)	S = 0.158 *p* = 0.169	S = −0.246 *p* = 0.065	S = −0.365 *p* = 0.011 **	S = −0.262 *p* = 0.053	S = −0.131 *p* = 0.214	S = 0.116 *p* = 0.240	S = −0.157 *p* = 0.169	S = −0.114 *p* = 0.268	S = 0.015 *p* = 0.463	S = 0.036 *p* = 0.413	S = 0.066 *p* = 0.351	S = −0.074 *p* = 0.328	S = 0.033 *p* = 0.421	S = 0.234 *p* = 0.076	S = −0.21 *p* = 0.092	S = 0.345 *p* = 0.016 **	S = 0.389 *p* = 0.007 **	S = 0.29 *p* = 0.036 *	-
Quality of life (environment)	P = 0.167 *p* = 0.155	P = −0.453 *p* = 0.002 **	P = −0.377 *p* = 0.009 **	P = −0.365 *p* = 0.011 **	P = −0.190 *p* = 0.124	P = −0.139 *p* = 0.200	P = −0.314 *p* = 0.026 *	P = −0.204 *p* = 0.132	P = −0.148 *p* = 0.185	P = −0.281 *p* = 0.042 *	P = −0.159 *p* = 0.167	P = −0.253 *p* = 0.060	P = −0.201 *p* = 0.110	P = −0.059 *p* = 0.362	P = −0.040 *p* = 0.401	P = 0.404 *p* = 0.005 **	P = 0.412 *p* = 0.005 **	P = 0.29 *p* = 0.037 *	P = 0.285 *p* = 0.039 *

P: r of Pearson; S: r of Spearman; *p*: <0.05. * Low correlation: P or S < 0.33. ** Moderate correlation: P or S 0.34 to 0.66. *** High correlation: P or S > 0.66. Questionnaire on Knowledge of Sexually Transmitted Diseases (STD-KQ).

## Data Availability

The datasets generated and/or analyzed during the current study are presented in this article.

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
