# Peer review of "Aspects of Sexuality in Quilombola Communities’ Young Adults in Eastern Brazilian Amazon"

_behavsci, 2023, doi:10.3390/bs13060492_

Round 1

Reviewer 2 Report

This is an interesting paper on an important topic. It would be helpful to provide more background on each of the measures used in the study. An additional sentence or two with more details on each survey would be enough to clarify for those unfamiliar with one of the measures used.

Author Response

Thank you for your thoughts! We have improved our description of the instruments and hope we have met this need. Please refer to 2.6. Data Collection starting in the second paragraph.

Reviewer 3 Report

The study examines the correlation between sexuality and and quality of life among young adults in Quilombola. However, below are my observation;

1. what kind of probability sampling technique was used to select the respondents

2. was the responses balance? 40/40 or was is unequal 25/50. can the result be generalize?

3. consider reframing the topic as it doesnt reflect the content

4. The introductory section is clumsy and not explanatory enough to highlight the aspect of sexuality that this study focused on

5. In the "Abstract" you stated that this study is the first to address this issue, while in paragraph 7, you stated that a few study has discussed the issue. Please align and proofread

6. How did you select male and female in the inclusion criteria. Since we have inclusion criteria, there was no reason for stating exclusion criteria. This is because inclusion criteria denote that anything not included, is automatically excluded.

7. What is FSFI, IIEF, CI etc in the result section. Kindly state the full meaning the first time it is used before using acronym. Do no assume that every ready must have known the meaning

8. What aspect of the qualitative study was conducted, and analysed. Who did you conduct the qualitative study with--who were the respondents/participant. How were they selected. Why were they selected? Are the respondents in the qualitative study the same as those in the quantitative study? How was the qualitative data analysed and synchronize in the result section? How was bias prevented or avoided?

9. How was "Knowledge" of sexuality measured?

10. What gap in knowledge or lacuna was filled by this study?

11. Who were those involve in coding the qualitative study and what was their expertise

Round 2

Reviewer 3 Report

I understand your busy nature to critically look at this manuscript and align your thought, especially in the method section. However, I advise you to look at it again as there must be synergy between the background of the study and the methods used. Points earlier highlighted has not been addressed

Author Response

Thank you for your consideration. We are seeking to align, according to your approval, our previous item with the method. In addition to changes specified in the previous clarified review attached to the first round, please review our changes to the Introduction item. We hope we have met your request. We are available for further clarifications.